# Relational Attention: Generalizing Transformers for Graph-Structured Tasks

**Cameron Diao** *
Department of Computer Science
Rice University
cwd2@rice.edu

**Ricky Loynd**
Microsoft Research
riloynd@microsoft.com

## Abstract

Transformers flexibly operate over sets of real-valued vectors representing task-specific entities and their attributes, where each vector might encode one word-piece token and its position in a sequence, or some piece of information that carries no position at all. But as set processors, standard transformers are at a disadvantage in reasoning over more general graph-structured data where nodes represent entities and edges represent relations *between* entities. To address this shortcoming, we generalize transformer attention to consider and update edge vectors in each transformer layer. We evaluate this *relational transformer* on a diverse array of graph-structured tasks, including the large and challenging CLRS Algorithmic Reasoning Benchmark. There, it dramatically outperforms state-of-the-art graph neural networks expressly designed to reason over graph-structured data. Our analysis demonstrates that these gains are attributable to relational attention's inherent ability to leverage the greater expressivity of graphs over sets.

## 1 Introduction

Graph-structured problems turn up in many domains, including knowledge bases (Hu et al., 2021; Bordes et al., 2013), communication networks (Leskovec et al., 2010), citation networks (McCallum et al., 2000), and molecules (Debnath et al., 1991; Zhang et al., 2020b). One example is predicting the bioactive properties of a molecule, where the atoms of the molecule are the nodes of the graph and the bonds are the edges. Along with their ubiquity, graph-structured problems vary widely in difficulty. For example, certain graph problems can be solved with a simple multi-layer perceptron, while others are quite challenging and require explicit modeling of relational characteristics.

Graph Neural Networks (GNNs) are designed to process graph-structured data, including the graph's (possibly directed) edge structure and (in some cases) features associated with the edges. In particular, they learn to represent graph features by passing messages between neighboring nodes and edges, and updating the node and (optionally) edge vectors. Importantly, GNNs typically restrict message passing to operate over the edges in the graph.

Figure 1: The relational transformer (RT) outperforms baseline GNNs on a set of 30 distinct graph-structured tasks from CLRS-30, averaged by algorithm class.

Standard transformers lack the relational inductive biases (Battaglia et al., 2018) that are explicitly built into the most commonly used GNNs. Instead, the transformer fundamentally consumes *unordered sets* of real-valued vectors, injecting no other assumptions. This allows entities carrying domain-specific attributes (like position) to be encoded as vectors for input to the same transformer architecture applied to different domains. Transformers have produced impressive results in a wide

---

*Work was done during an internship at Microsoft Research.

| One layer of computation | Edge Vector Incorporation | Transformer Machinery | | |
| --- | --- | --- | --- | --- |
| | | None | Some | Full |
|  | Round-trip node-edge communication | Some GNNs | A few models | *RT: Relational Transformer* |
|  | Restricted node-edge communication | *Most common GNNs* | Some GNNs and transformer variants | Some transformer variants |
|  | No edge vectors | — | — | *Standard transformer* |

Figure 2: Categories of GNNs and Transformers, compared in terms of transformer machinery and edge vector incorporation. Model categories tested in our experiments are marked in bold.

variety of domains, starting with machine translation (Vaswani et al., 2017), then quickly impacting language modeling (Devlin et al., 2019) and text generation (Brown et al., 2020). They are revolutionizing image processing (Dosovitskiy et al., 2021) and are being applied to a growing variety of settings including reinforcement learning (RL), both online (Loynd et al., 2020; Parisotto et al., 2020) and offline (Chen et al., 2021; Janner et al., 2021).

Many of the domains transformers succeed in consist of array-structured data, such as text or images. By contrast, graph data is centrally concerned with pairwise relations between entities, represented as edges and edge attributes. Graphs are more general and expressive than sets, in the sense that a set is a special case of a graph—one without edges. So it is not immediately obvious how graph data can be processed by transformers in a way that preserves relational information. Transformers have been successfully applied to graph-structured tasks in one of two broad ways. Certain works, most recently TokenGT (Kim et al., 2022), encode graphs as sets of real-valued vectors passed to a standard transformer. Other works change the transformer architecture itself to consider relational information, e.g. by introducing relative position vectors to transformer attention. We discuss these and many other such approaches in Section 4.

**Our novel contribution is relational attention, a mathematically elegant extension of transformer attention, which incorporates edge vectors as first-class model components.** We call the resulting transformer architecture the Relational Transformer (RT). As a native graph-to-graph model, RT does not rely on special encoding schemes to input or output graph data.

We find that RT outperforms baseline GNNs on a large and diverse set of difficult graph-structured tasks. In particular, RT establishes dramatically improved state-of-the-art performance (Figure 1) over baseline GNNs on the challenging CLRS-30 (Veličković et al., 2022), which comprises 30 different algorithmic tasks in a framework for probing the reasoning abilities of graph-to-graph models. To summarize our main contributions:

- We introduce the relational transformer for application to arbitrary graph-structured tasks, and make the implementation available at `https://github.com/CameronDiao/relational-transformer`.
- We evaluate the reasoning power of RT on a wide range of challenging graph-structured tasks, achieving new state-of-the-art results on CLRS-30.
- We enhance the CLRS-30 framework to support evaluation of a broader array of models (Section 5.1.2).
- We improve the performance of CLRS-30 baseline models by adding multi-layer functionality, and tuning their hyperparameters (Section 5.1.1).

## 2 GRAPH NEURAL NETWORKS

We introduce the graph-to-graph model formalism used in the rest of this paper, inspired by Battaglia et al. (2018). The input graph is a directed, attributed graph $G = (\mathcal{N}, \mathcal{E})$, where $\mathcal{N}$ is an unordered set of node vectors $\mathbf{n}_i \in \mathbb{R}^{d_n}$, $i$ denoting the $i$-th node. $\mathcal{E}$ is a set of edge vectors $\mathbf{e}_{ij} \in \mathbb{R}^{d_e}$, where directed edge $(i, j)$ points from node $j$ to node $i$. Each layer $l$ in the model accepts a graph $G^l$ as

input, processes the graph's features, then outputs graph $G^{l+1}$ with the same structure as $G^l$, but with potentially updated node and edge vectors. Certain tasks may also include a single global vector with the input or output graph, but we omit those details from our formalism since they are not what distinguishes the approaches described below.

Each layer $l$ of a graph-to-graph model is comprised of two update functions $\phi$ and an aggregation function $\bigoplus$,

$$\text{Node vector } \mathbf{n}_i^{l+1} = \phi_n\left(\mathbf{n}_i^l, \bigoplus_{j \in \mathcal{L}_i} \psi^m\left(\mathbf{e}_{ij}^l, \mathbf{n}_i^l, \mathbf{n}_j^l\right)\right) \tag{1}$$

$$\text{Edge vector } \mathbf{e}_{ij}^{l+1} = \phi_e\left(\mathbf{e}_{ij}^l, \mathbf{e}_{ji}^l, \mathbf{n}_i^{l+1}, \mathbf{n}_j^{l+1}\right) \tag{2}$$

where $\mathcal{L}_i$ denotes the set of node $i$'s neighbors (optionally including node $i$), and $\psi^m$ denotes a message function. The baseline GNNs listed below let $\phi_e$ be the identity function such that $\mathbf{e}_{ij}^{l+1} = \mathbf{e}_{ij}^l$ for all edges $(i, j)$. Furthermore, they use a permutation-invariant aggregation function for $\bigoplus$. The following details are from Veličković et al. (2022):

In **Deep Sets** (Zaheer et al., 2017), the only edges are self-connections so each $\mathcal{L}_i$ is a singleton set containing node $i$.

In **Graph Attention Networks (GAT)** (Veličković et al., 2018; Brody et al., 2022), $\bigoplus$ is self-attention, and the message function $\psi^m$ merely extracts the sender features: $\psi^m(\mathbf{e}_{ij}^l, \mathbf{n}_i^l, \mathbf{n}_j^l) = \mathbf{W}_m \mathbf{n}_j^l$, where $\mathbf{W}_m$ is a weight matrix.

In **Message Passing Neural Networks (MPNN)** (Gilmer et al., 2017), edges lie between any pair of nodes and $\bigoplus$ is the max pooling operation.

In **Pointer Graph Networks (PGN)** (Veličković et al., 2020), edges are constrained by the adjacency matrix and $\bigoplus$ is the max pooling operation.

## 3 Relational Transformer

We aim to design a mathematically elegant extension of transformer attention, which incorporates edge vectors as first-class model components. This goal leads us to the following design criteria:

1. Preserve all of the transformer's original machinery (though still not fully understood), for its empirically established advantages.
2. Introduce directed edge vectors to represent relations between entities.
3. Condition transformer attention on the edge vectors.
4. Extend the transformer layer to consume edge vectors and produce updated edge vectors.
5. Preserve the transformer's $\mathcal{O}\left(N^2\right)$ computational complexity.

### 3.1 Relational Attention

(See Appendix A for a mathematical overview of transformers.) In addition to accepting node vectors representing entity features (as do all transformers), RT also accepts edge vectors representing relation features, which may include edge-presence flags from an adjacency matrix. But RT operates over a fully connected graph, unconstrained by any input adjacency matrix.

Transformer attention projects QKV vectors from each node vector, then computes a dot-product between each pair of vectors $\mathbf{q}_i$ and $\mathbf{k}_j$. This dot-product determines the degree to which node $i$ attends to node $j$. Relational attention's central innovation (illustrated in Figure 3) is to condition the QKV vectors on the directed edge $\mathbf{e}_{ij}$ between the nodes, by concatenating that edge vector with each node vector prior to the linear transformations:

$$\mathbf{q}_{ij} = [\mathbf{n}_i, \mathbf{e}_{ij}]\mathbf{W}^Q \qquad \mathbf{k}_{ij} = [\mathbf{n}_j, \mathbf{e}_{ij}]\mathbf{W}^K \qquad \mathbf{v}_{ij} = [\mathbf{n}_j, \mathbf{e}_{ij}]\mathbf{W}^V \tag{3}$$

where each weight matrix $\mathbf{W}$ is now of size $\mathbb{R}^{(d_n + d_e) \times d_n}$, and $d_e$ is the edge vector size. To implement this efficiently and exactly, we split each weight matrix $\mathbf{W}$ into two separate matrices for

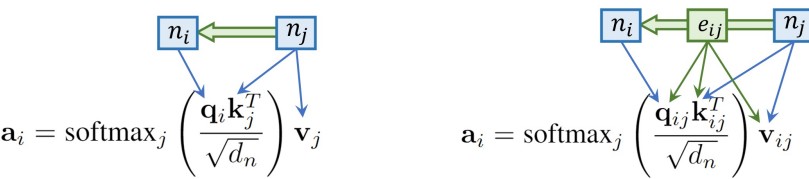

$$\mathbf{a}_i = \mathrm{softmax}_j \left( \frac{\mathbf{q}_i \mathbf{k}_j^T}{\sqrt{d_n}} \right) \mathbf{v}_j \qquad\qquad \mathbf{a}_i = \mathrm{softmax}_j \left( \frac{\mathbf{q}_{ij} \mathbf{k}_{ij}^T}{\sqrt{d_n}} \right) \mathbf{v}_{ij}$$

Figure 3: (Left) Standard transformer attention conditions the QKV computation on node vectors. (Right) Relational attention conditions this computation on the intervening edge vector as well.

projecting node and edge vectors, project the edge vector to three embeddings, then add those to the node's usual attention vectors:

$$\mathbf{q}_{ij} = \left( \mathbf{n}_i \mathbf{W}_n^Q + \mathbf{e}_{ij} \mathbf{W}_e^Q \right) \qquad \mathbf{k}_{ij} = \left( \mathbf{n}_j \mathbf{W}_n^K + \mathbf{e}_{ij} \mathbf{W}_e^K \right) \qquad \mathbf{v}_{ij} = \left( \mathbf{n}_j \mathbf{W}_n^V + \mathbf{e}_{ij} \mathbf{W}_e^V \right) \quad (4)$$

While we have described relational attention in terms of fully connected self-attention, it applies equally to restricted forms of attention such as causal attention, cross-attention, or even restricted GAT-like attention that passes messages only over the edges present in a graph's adjacency matrix. Relational attention is compatible with multi-head attention, and leaves the transformer's $\mathcal{O}\left(N^2\right)$ complexity unchanged. Our implementation maintains the high GPU utilization that makes transformers efficient.

### 3.2 Edge Updates

To update edge vectors in each layer of processing, we follow the general pattern used by transformers to update node vectors: first aggregate messages into one, then use the result to perform a local update. In self-attention each node attends to all nodes in the graph. But having each of the $N^2$ edges attend to $N$ nodes would raise computational complexity to $\mathcal{O}\left(N^3\right)$. Instead, we restrict the edge's aggregation function to gather messages only from its immediate locale, which consists of its two adjoining nodes, itself, and the directed edge running in the opposite direction:

$$\mathbf{e}_{ij}^{l+1} = \phi_e \left( \mathbf{e}_{ij}^l, \mathbf{e}_{ji}^l, \mathbf{n}_i^{l+1}, \mathbf{n}_j^{l+1} \right) \tag{5}$$

We compute the aggregated message $\mathbf{m}_{ij}^l$ by first concatenating the four neighbor vectors, then applying a single linear transformation to the concatenated vector, followed by a ReLU non-linearity:

$$\mathbf{m}_{ij}^l = \mathrm{ReLU} \left( \mathrm{concat} \left( \mathbf{e}_{ij}^l, \mathbf{e}_{ji}^l, \mathbf{n}_i^{l+1}, \mathbf{n}_j^{l+1} \right) \mathbf{W}_4 \right) \tag{6}$$

where $\mathbf{W}_4 \in \mathbb{R}^{(2d_e + 2d_n) \times d_{eh1}}$, and the non-linear ReLU operation takes the place of the non-linear softmax in regular attention.

The remainder of the edge update function is essentially identical to the transformer node update function:

$$\mathbf{u}_{ij}^l = \mathrm{LayerNorm} \left( \mathbf{m}_{ij}^l \mathbf{W}_5 + \mathbf{e}_{ij}^l \right) \qquad \mathbf{e}_{ij}^{l+1} = \mathrm{LayerNorm} \left( \mathrm{ReLU}(\mathbf{u}_{ij}^l \mathbf{W}_6) \mathbf{W}_7 + \mathbf{u}_{ij}^l \right) \tag{7}$$

where $\mathbf{W}_5 \in \mathbb{R}^{d_{eh1} \times d_e}$, $\mathbf{W}_6 \in \mathbb{R}^{d_e \times d_{eh2}}$, $\mathbf{W}_7 \in \mathbb{R}^{d_{eh2} \times d_e}$, and $d_{eh1}$ and $d_{eh2}$ are the hidden layer sizes of the edge feed-forward networks. Node vectors are updated before edge vectors within each RT layer, so that a node aggregates information from the entire graph before its adjoining edges use that information in their local updates. This is another instance of the aggregate-then-update pattern employed by both transformers and GNNs for node vectors.

## 4 Prior Work

We categorize, then discuss prior works based on their use of transformer machinery. We further divide each category based on edge vector incorporation, highlighting works that use full node-edge round-tripping, here defined as the process in which edge vectors directly condition node updates and node vectors directly condition edge updates. See Figure 2 for visual comparisons.

**Encoding Graphs to Sets** For clarity, we define the transformer architecture to exclude any modules that encode inputs to the transformer or decode its outputs. Several prior works have made progress on the challenge of applying standard transformers (Vaswani et al., 2017) to graph-structured tasks by representing graphs as sets of tokens, e.g. with positional encodings. TokenGT (Kim et al., 2022), for example, treats all nodes and edges of a graph as independent tokens, augmented with token-wise embeddings. Graphormer (Ying et al., 2021) and Graph-BERT (Zhang et al., 2020a) introduce structural encodings that are applied to each node prior to transformer processing. GraphTrans (Wu et al., 2021) and ReFormer (Yang et al., 2022) perform initial convolutions or message passing before the transformer module.

**Relative Position Vectors** A number of prior works have modified self-attention to implement relative positional encodings (Ying et al., 2021; Cai & Lam, 2020b; Shaw et al., 2018; Hellendoorn et al., 2020; Dai et al., 2019). To compare these formulations with RT, we expand relational attention's dot-product into four terms as follows:

$$\mathbf{q}_{ij}\mathbf{k}_{ij}^\top = \left([\mathbf{n}_i, \mathbf{e}_{ij}]\mathbf{W}^Q\right)\left([\mathbf{n}_j, \mathbf{e}_{ij}]\mathbf{W}^K\right)^\top \tag{8}$$

$$= \left(\mathbf{n}_i\mathbf{W}_n^Q + \mathbf{e}_{ij}\mathbf{W}_e^Q\right)\left(\mathbf{n}_j\mathbf{W}_n^K + \mathbf{e}_{ij}\mathbf{W}_e^K\right)^\top \tag{9}$$

$$= \left(\mathbf{n}_i\mathbf{W}_n^Q + \mathbf{e}_{ij}\mathbf{W}_e^Q\right)\left(\left(\mathbf{n}_j\mathbf{W}_n^K\right)^\top + \left(\mathbf{e}_{ij}\mathbf{W}_e^K\right)^\top\right) \tag{10}$$

$$= \mathbf{n}_i\mathbf{W}_n^Q\left(\mathbf{n}_j\mathbf{W}_n^K\right)^\top + \mathbf{n}_i\mathbf{W}_n^Q\left(\mathbf{e}_{ij}\mathbf{W}_e^K\right)^\top + \mathbf{e}_{ij}\mathbf{W}_e^Q\left(\mathbf{n}_j\mathbf{W}_n^K\right)^\top + \mathbf{e}_{ij}\mathbf{W}_e^Q\left(\mathbf{e}_{ij}\mathbf{W}_e^K\right)^\top \tag{11}$$

The transformer of Vaswani et al. (2017) employs only the first term. The transformer of Shaw et al. (2018) adds part of the second term, leaving out one weight matrix $\mathbf{W}_e^K$, and GREAT (Hellendoorn et al., 2020) adds the entire third term. The Transformer-XL (Dai et al., 2019) and Graph Transformer of Cai & Lam (2020b) use parts of all four terms, but leave out two of the eight weight matrices $\mathbf{W}_e^Q$ and $\mathbf{W}_e^K$. Each work above uses edge vectors only for relative positional information. RT employs all four terms, allows the edge vectors to represent arbitrary information depending on the task, and updates the edge vectors in each layer of computation.

**Transformers With Restricted Node-Edge Communication** A few prior graph-to-graph transformers restrict round-trip communication between nodes and edges. EGT (Hussain et al., 2022) does a form of node-edge round-tripping but introduces single-scalar bottlenecks (per-edge, per-head). The Graph Transformer of Dwivedi & Bresson (2021), SAT (Chen et al., 2022), and SAN (Kreuzer et al., 2021) condition node attention coefficients on edge vectors, but they do not explicitly condition node vector updates on edge vectors. GRPE (Cai & Lam, 2020a) also conditions node attention on edge vectors, e.g. by adding edge vectors to the node value vectors. But the edge vectors themselves are not explicitly updated using node vectors.

**Highly Modified Transformers** Bergen et al. (2021) introduce the Edge Transformer which replaces standard transformer attention with a triangular attention mechanism that takes edge vectors into account and updates the edge vectors in each layer. This differs from RT in three important respects. First, triangular attention is a completely novel form of attention, unlike relational attention which is framed as a natural extension of standard transformer attention. Second, triangular attention ignores node vectors altogether, and thereby requires node input features and node output predictions to be somehow mapped onto edges. And third, triangular attention's computational complexity is $\mathcal{O}\left(N^3\right)$ in the number of nodes, unlike RT's relational attention which maintains the $\mathcal{O}\left(N^2\right)$ transformer complexity. Like Edge Transformer, Nodeformer (Wu et al., 2022) employs a novel form of attention, but in this case with $\mathcal{O}\left(N\right)$ complexity. Nodeformer does not perform node-edge round-tripping, and introduces single-scalar bottlenecks per-edge. Another transformer, GraphGPS (Rampasek et al., 2022), is described by the authors as an MPNN+Transformer hybrid, which does support full node-edge round-tripping. Unlike RT, GraphGPS represents a significant departure from the standard transformer architecture.

Finally, attentional GNNs, such as GAT (Veličković et al., 2018), GATv2 (Brody et al., 2022), Edgeformer (Jin et al., 2023), kgTransformer (Liu et al., 2022), Relational Graph Transformer (Feng et al., 2022), HGT (Hu et al., 2020), and Simple-HGN (Lv et al., 2021) aggregate features across neighborhoods based on transformer-style attention coefficients. However, unlike transformers, attentional GNNs only compute attention over input edge vectors, and (except in Edgeformer and kgTransformer) the edge vectors are not updated in each layer. In particular, kgTransformer, Relational

Graph Transformer, HGT, and Simple-HGN modify transformer attention to consider hetereogeneous structures in the graph data, such that the model can differentiate between types of nodes and edges.

**Other GNNs** MPNN (Gilmer et al., 2017) is a popular GNN that accepts entire edge vectors as input, as do some other works such as MXMNet (Zhang et al., 2020c) and G-MPNN (Yadati, 2020). But apart from EGNN (Gong & Cheng, 2019) and Censnet (Jiang et al., 2019), relatively few GNNs update the edge vectors themselves. None of these GNNs use the full transformer machinery, and in general many GNNs are designed for specific settings, such as quantum chemistry.

Unlike any of these prior works, RT preserves all of the original transformer machinery, while adding full bidirectional conditioning of node and edge vector updates.

## 5 EXPERIMENTS

We evaluate RT against common GNNs on the diverse set of graph-structured tasks provided by CLRS-30 (Veličković et al., 2022), which was designed to measure the reasoning abilities of neural networks. This is a common motivation for tasking neural networks to execute algorithms (Zaremba & Sutskever, 2014; Kaiser & Sutskever, 2015; Trask et al., 2018; Kool et al., 2019). RT outperforms baseline GNNs by wide margins, especially on tasks that require processing of node relations (Section 5.1.5). We further evaluate RT against GNNs on the end-to-end shortest paths task provided by Tang et al. (2020), where again RT outperforms the baselines (Section 5.2). Our final experiment (Appendix B) evaluates RT against a standard transformer on a reinforcement learning task where *no graph structure is provided*. We find that RT decreases error rates of the RL agent significantly.

### 5.1 STEP-BY-STEP REASONING

In CLRS-30, each step in a task is framed as a graph-to-graph problem, even for algorithms that may seem unrelated to graphs. To give an example, for list sorting algorithms, each input list element is treated as a separate node and predecessor links are added to order the elements. Task data is organized into task inputs, task outputs, and 'hints', which are intermediate inputs and outputs for the intervening steps of an algorithm. Data is comprised of combinations of node, edge, and/or global features, which can be of five possible types: scalars, categoricals, masks, binary masks, or pointers.

CLRS-30 employs an encode-process-decode framework for model evaluation. Input features are encoded using linear layers, then passed to the model (called a processor) being tested. The model performs one step of computation on the inputs, then it outputs a set of node vectors, which are passed back to the model as input on the next step. On each step, the model's output node vectors are decoded by the framework (using linear layers) and compared with the targets (either hints or final outputs) to compute training losses or evaluation scores.

Certain CLRS-30 tasks provide a global vector with each input graph. As per CLRS-30 specifications, the baseline GNNs handle global vectors by including them as messages in each update step. They do not propagate global vectors through the steps of the algorithm. RT can use two different methods for handling global vectors and we evaluate both in Section 5.1.4.

### 5.1.1 BASELINE GNNS

We began by reproducing the published results of key baseline models on the eight representative tasks (one per algorithm class) listed in Figure 3 of Veličković et al. (2022) and in our Table 13. For several of the following experiments, we refer to these as the *8 core tasks*. Our results on these tasks agree closely with the published CLRS-30 results (Table 11). See Appendix D for details of our train/test protocol. We chose not to include Memnet in our experiments given our focus on standard GNNs, and given Memnet's poor performance in the original CLRS-30 experiments. Missing details[1] made it impossible to reproduce the published GAT results on CLRS-30.

The published CLRS-30 results show sharp drops in out-of-distribution (OOD) performance for all models. For instance, MPNN's average evaluation score drops from 96.63% on the validation set to 51.02% on the test set. We note that small training datasets can induce overfitting even in models that are otherwise capable of generalizing to OOD test sets. To mitigate this spurious form of overfitting,

---

[1]See the final comment in https://github.com/deepmind/clrs/issues/92

Table 1: Mean test scores of all tuned models on the eight CLRS-30 algorithm classes.

| Class | Deep Sets | GAT-v1 | GAT-v2 | MPNN | PGN-u | PGN-m | PGN-c | RT (Ours) |
|---|---|---|---|---|---|---|---|---|
| Divide & Conquer | 12.29% | 15.19% | 14.80% | 16.14% | 16.71% | 51.30% | 51.30% | **66.52%** |
| Dynamic Programming | 68.29% | 63.88% | 59.22% | 68.81% | 68.56% | 71.07% | 71.13% | **83.20%** |
| Geometry | 65.47% | 68.94% | 67.80% | 83.03% | 67.77% | 66.63% | 67.78% | **84.55%** |
| Graphs | 42.09% | 52.79% | 55.53% | 63.30% | 59.16% | 64.36% | 64.59% | **65.33%** |
| Greedy | 77.83% | 75.75% | 75.03% | **89.40%** | 75.30% | 76.72% | 76.72% | 85.32% |
| Search | 50.99% | 35.37% | 38.04% | 43.94% | 50.98% | 54.21% | 60.39% | **65.31%** |
| Sorting | **68.89%** | 21.25% | 17.82% | 27.12% | 28.93% | 2.48% | 28.93% | 50.01% |
| Strings | 2.92% | 1.36% | 1.57% | 2.09% | 1.61% | 1.17% | 1.82% | **32.52%** |
| Average | 48.60% | 41.82% | 41.23% | 49.23% | 46.13% | 48.49% | 52.83% | **66.60%** |

we expanded the training datasets by 10x to 10,000 examples generated from the same canonical random seed of 1, and evaluated the effects on the 8 core tasks. As shown in Table 9, expanding the training set significantly boosts the performance of all baseline models. For all of the other experiments in this work, we use the larger training sets of 10,000 examples.

Following Veličković et al. (2022), we compute results for two separate PGN models, masked (PGN-m) and unmasked (PGN-u), then select the best result on each task to compute the average shown for the combination PGN-c model (which is called PGN in the CLRS-30 results). Note therefore that PGN-c does not represent a single model. But it does represent the performance that would be achievable by a PGN model that adaptively learned when to use masking.

We found in early experiments that RT obtained far better results than those of the CLRS-30 baseline GNNs. So to further enhance the performance of the baseline GNNs, we extended them to support multiple layers (rounds of message passing) per algorithmic step, and thoroughly tuned their hyperparameters (see Appendix C). This significantly improved the baseline results (see Table 10).

See Table 12 for results comparing CLRS-30 baseline model performances with and without our proposed changes. On certain tasks, baseline score variance increased along with mean scores. For example, on the Jarvis' March task, tuning raised the score of MPNN from $22.99 \pm 3.87\%$ to $59.31 \pm 29.3\%$. See Appendix G for detailed analysis of score variance.

### 5.1.2 ENHANCEMENTS OF CLRS-30'S FRAMEWORK

Most GNNs consume and produce node vectors, and many also consume edge vectors or edge types. However, relatively few GNNs (and none of the CLRS-30 baseline models) are designed to output edge vectors. Because of this, the CLRS-30 framework does not support edge vector outputs from a processor network. To test models such as RT that have these abilities, we extended the CLRS-30 framework to accept edge and global vectors from the processor at each step, and pass these vectors back to the processor as input on the next step. The framework handles node vectors as usual.

In framing algorithms as graph-to-graph tasks, CLRS-30 relies heavily on what it terms a *node pointer*, which is conceptually equivalent to a directed edge pointing from one node to another. Since the CLRS-30 baseline models do not output edge vectors, a decoder in the CLRS-30 framework uses the model's output node vectors to create node pointers. But for models like RT that output edge vectors, it is more natural to decode node pointers from those edge vectors alone. To better support such models, we added a flag to enable this modified behavior in the CLRS-30 framework.

### 5.1.3 MAIN RESULTS

After tuning hyperparameters for all models (Appendix C), we evaluated RT against the six baseline GNNs on all CLRS-30 tasks, using 20 seeds. The full results are presented in Table 19. RT outperforms the top-scoring baseline model (MPNN) by **11%** overall. As bolded in the table, RT scores the highest on **11 out of 30** tasks. RT is also the best-performing model on **6 of 8** algorithmic classes (Table 1), and scores the highest when results are averaged over those classes (Figure 1). See Table 8 for the algorithm-class mappings. For convenience, Figure 1 includes the prior results (labeled as MPNN-pr and DeepSets-pr) for single-layer GNNs from Veličković et al. (2022). In summary, RT significantly outperforms all baseline GNN models over the CLRS-30 tasks.

### 5.1.4 ABLATIONS

Using only the 8 core tasks (except where noted), we perform several ablations to analyze the factors behind RT's solid performance on CLRS-30.

**Transformer** - We compare RT to a standard, set-based transformer (Vaswani et al., 2017) by disabling edge vectors and features in RT. Table 13 shows that performance collapses by almost 40% without edge vectors and relational attention, even after re-tuning its hyperparameters.

**Layers** - The tuned RT uses three layers of computation per algorithmic step. When restricted to a single layer, performance drops drastically (Table 14), even after re-tuning the other hyperparameters. However, single-layer RT still outperforms the top-scoring MPNN by **10.69%** on the 8 core tasks, suggesting that relational attention improves expressivity even when restricted to a single layer of computation.

**Global vector** - Many CLRS-30 tasks provide a global feature vector as input to the processor model. We designed RT to handle this global vector by either concatenating it to each input node vector, or by passing it to a dedicated core node (Loynd et al., 2020; Guo et al., 2019). Hyperparameter tuning chose concatenation instead of the core node option, so concatenation was used in all experiments. But in this ablation, the core-node method obtained slightly higher test scores on 7 tasks that use global vectors as inputs to the processor (Table 16). The score difference of 0.08% was marginal, providing no empirical basis to prefer one method of handling the global vector over the other.

**Node pointer decoding** - We assess impact of the flag we added to the CLRS-30 framework, which can be used to decode node pointers from edge vectors only. Compared to using the original decoding procedure, using the flag improved performance by a small amount (0.30%) (Table 15).

**Disabling edge updates** - We disable edge updates in RT such that RT relies solely on relational attention to process input features. Table 17 shows the resulting drop in performance, from **81.30%** to **53.99%**. This indicates that edge updates are crucial for RT's learning of relational characteristics in the graph data. As a final note, RT without edge updates still outperforms the transformer by **11.65%**, demonstrating the effectiveness of relational attention even without updated edges.

### 5.1.5 ALGORITHMIC ANALYSIS

We investigate the reasoning power of RT based on its test performance on specific algorithmic tasks. We only provide possible explanations here, in line with previous work (Veličković et al., 2022; Veličković et al., 2020; Xu et al., 2020).

**Underperformance** - The greedy class is one of two where RT is outperformed (by just one other model). The two greedy tasks, activity selector and task scheduling, require selecting node entities that minimize some metric at each step. For example, in task scheduling, the optimal solution involves repeatedly selecting the task with the smallest processing time. The selection step is aligned with max pooling in GNNs: Veličković et al. (2020) demonstrate how max pooling aligns with making discrete decisions over neighborhoods. Here, each neighborhood represents a set of candidate entities to be selected from. MPNN, PGN-u, and PGN-m all perform max pooling at each step of message passing. On the other hand, RT performs soft attention pooling, which does not align with the discrete decision-making required to execute greedy algorithms. This may explain RT's underperformance on activity selector and task scheduling, as well as Prim's and Kruskal's.

**Overperformance** - RT overwhelmingly beats baseline GNNs on dynamic programming (DP) tasks. This is surprising, considering that GNNs have been proven to align well with dynamic programming routines (Dudzik & Veličković, 2022). To explain RT's overperformance, we consider 1) edge updates and 2) relational attention. For 1), Ibarz et al. (2022) observe that several algorithms in CLRS-30, especially those categorized as DP, require edge-based reasoning—where edges store values, and update those values based on other edges' values. These algorithms do not use node representations in their update functions, yet the baseline GNNs can only learn these update functions using message passing between node representations. On the other hand, RT directly supports edge-based reasoning by representing and updating edges. The hypothesis that RT actually uses this ability is supported by the fact that RT beats baseline GNNs on most of the 6 edge-centric tasks (Find Maximum Subarray, Insertion Sort, Matrix Chain Order, and Optimal BST), though not on the other 2 (Dijkstra and Floyd-Warshall). For 2), recall from Section 3.1 that relational attention is an extension of standard

transformer attention. Standard attention itself is a specific instance of the message-passing function described in Dudzik & Veličković (2022), which is part of the author's framework for aligning GNNs and DP routines. To see how this is the case, the reader can compare our equation equation 12 with equation 1 in Dudzik & Veličković (2022). From this comparison, and from 1), we expect RT to perform well on both of the edge-centric DP tasks Matrix Chain Order and Optimal BST. We find that RT obtains best scores on both of them, by 5.24% and 3.66% respectively.

## 5.2 END-TO-END ALGORITHMIC REASONING

CLRS-30 evaluates reasoning ability by examining how closely a model can emulate each step of an algorithm. But we may also evaluate reasoning ability by training a model to execute an algorithm end-to-end (Xu et al., 2020). We use the task provided by Tang et al. (2020) to evaluate RT in this way. Specifically, we task RT with finding a shortest path distance between two nodes in an undirected lobster graph. The node features are one-hot encoded for source, destination, and remaining. The edge features are binary presence values.

**Main Results** We use the same experiment settings as Tang et al. (2020) for the shortest path task. Importantly, we use their method for testing graph size generalizability by training models on graphs of size $[4, 34]$ and evaluating models on graphs of size 100. Furthermore, all models use 30 layers. Results are reported using the relative loss metric introduced by the authors, defined as $|y-\hat{y}|/|y|$ given a label $y$ and a prediction $\hat{y}$. We compare RT to their two baselines, Graph Convolution Network or GCN (Kipf & Welling, 2017) and GAT (Veličković et al., 2018). Results are averaged across 20 random seeds. RT outperforms both baselines with an

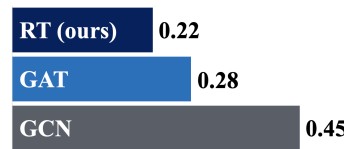

Figure 4: Average Relative Loss on Shortest Paths

average relative loss of **0.22**, compared to GCN's **0.45** and GAT's **0.28** (Figure 4). These results were obtained without using the iterative module proposed by Tang et al. (2020) that introduces a stopping criterion to message passing computations.

## 6 CONCLUSION AND FUTURE WORK

We propose the relational transformer (RT), an elegant extension of the standard transformer to operate on graphs instead of sets. It incorporates edge information through relational attention in a principled and computationally efficient way. Our experimental results demonstrate that RT performs consistently well across a diverse range of graph-structured tasks. Specifically, RT outperforms baseline GNNs on CLRS-30 by wide margins, and also outperforms baseline models on end-to-end algorithmic reasoning. RT even boosts transformer performance on the Sokoban task, where graph structure is entirely hidden and must be discovered by the RL agent.

Beyond establishing RT's state-of-the-art results on CLRS-30, we enhance performance of the CLRS-30 baseline models, and contribute extensions to the CLRS-30 framework, broadening the scope of models that can be evaluated on the benchmark tasks. All of these improvements make CLRS-30 tasks and baselines more appealing for evaluating current and future models.

In general, comparing GNNs with transformer-based models like RT on common benchmarks is an important challenge for the community. We have made progress on that challenge by rigorously evaluating RT against standard baseline GNNs on the large and challenging CLRS-30 benchmark, but we leave experiments with other transformer-based approaches for future work. One difficulty is the fact that the CLRS-30 framework is written in Jax, and few if any of these transformers have Jax implementations available. But we have improved the CLRS benchmark itself to make such comparisons more practical in the future.

In future work, we aim to fully leverage the richness of CLRS-30 to more thoroughly investigate RT's capabilities. For example, recent extensions to the CLRS framework (Ibarz et al., 2022) allow us to task RT with executing several algorithms simultaneously, which requires knowledge transfer between algorithms. We also plan to evaluate RT on a wider range of real-world graph settings, such as the molecular domain using the large-scale QM9 dataset (Wu et al., 2018). Finally, we aim to relax the locality bottleneck of RT's edge updates by allowing edges to attend to other edges directly in a computationally efficient manner.

ACKNOWLEDGMENTS

We wish to thank our many collaborators for their valuable feedback, including Roland Fernandez, who also provided the XT ML tool that made research at this scale possible.

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

## A    TRANSFORMERS

We describe the transformer architecture introduced by Vaswani et al. (2017). This description also applies to most transformer variants proposed over the years. Layer superscripts are employed to distinguish input vectors from output vectors, and are often omitted for vectors inside the same layer.

Although the transformer is a set-to-set model, it can be described using our graph-to-graph formalism as limited to computation over *nodes* only. Each transformer layer is a function passing updated node vectors to the next layer. A single transformer layer can therefore be expressed as a modified version of equation 1:

$$\mathbf{n}_i^{l+1} = \phi_n \left( \mathbf{n}_i^l, \bigoplus_{j \in \mathcal{L}_i} a\left(\mathbf{n}_i^l, \mathbf{n}_j^l\right) \psi^m\left(\mathbf{n}_j^l\right) \right) \tag{12}$$

where $a\left(\mathbf{n}_i^l, \mathbf{n}_j^l\right)$ computes the attentional coefficient $\alpha_{ij}^l$ applied by node $i$ to the value vector $\mathbf{v}_j^l$, which is computed by $\psi^m\left(\mathbf{n}_j^l\right)$, a linear transformation:

$$\alpha_{ij}^l = a\left(\mathbf{n}_i^l, \mathbf{n}_j^l\right) \qquad \mathbf{v}_j^l = \psi^m\left(\mathbf{n}_j^l\right) = \mathbf{n}_j^l W^V \tag{13}$$

The attentional coefficients applied by node $i$ to the set of all nodes $j$ is a probability distribution $\mathbf{a}_i$ computed by the softmax function over a set of vector dot products:

$$\mathbf{a}_i = \text{softmax}_j \left( \frac{\mathbf{q}_i \mathbf{k}_j^T}{\sqrt{d_n}} \right) \tag{14}$$

The *QKV* vectors introduced above are linear transformations of node vectors:

$$\mathbf{q}_i = \mathbf{n}_i W^Q \qquad \mathbf{k}_j = \mathbf{n}_j W^K \qquad \mathbf{v}_j = \mathbf{n}_j W^V \tag{15}$$

where $\mathbf{W}^Q \in \mathbb{R}^{d_n \times d_n}$, $\mathbf{W}^K \in \mathbb{R}^{d_n \times d_n}$, $\mathbf{W}^V \in \mathbb{R}^{d_n \times d_n}$, and $d_n$ is the node vector size. These $\mathbf{W}$ matrices (like all other trainable parameters in $\phi_n$) are not shared between transformer layers.

The aggregation function $\bigoplus$ sums the incoming messages from all nodes $\mathcal{L}_i$ in the completely connected graph:

$$\mathbf{m}_i^l = \bigoplus_{j \in \mathcal{L}_i} a\left(\mathbf{n}_i^l, \mathbf{n}_j^l\right) \psi^m\left(\mathbf{n}_j^l\right) = \sum_j \alpha_{ij}^l \mathbf{v}_j^l \tag{16}$$

This aggregated message $\mathbf{m}_i^l$ is then passed to the local update function $\phi_n$ (shared by all nodes), which is the following stack of linear layers, skip connections, layer normalization and a ReLU activation function:

$$\mathbf{u}_i^l = \text{LayerNorm}\left(\mathbf{m}_i^l \mathbf{W}_1 + \mathbf{n}_i^l\right) \tag{17}$$

$$\mathbf{n}_i^{l+1} = \text{LayerNorm}\left(\text{ReLU}(\mathbf{u}_i^l \mathbf{W}_2)\mathbf{W}_3 + \mathbf{u}_i^l\right) \tag{18}$$

where $\mathbf{W}_1 \in \mathbb{R}^{d_n \times d_n}$, $\mathbf{W}_2 \in \mathbb{R}^{d_n \times d_{nh}}$, $\mathbf{W}_3 \in \mathbb{R}^{d_{nh} \times d_n}$, and $d_{nh}$ is the hidden layer size of the feed-forward network.

This overview of the transformer architecture has focused on the fully connected case of self-attention. For brevity we have omitted the details of multi-head attention, bias vectors, and the stacking of vectors into matrices for maximal GPU utilization.

## B    SOKOBAN EXPERIMENTS

In the experiments described in the main text, the model received edge feature vectors as inputs. The question we pose here is whether RT's latent edge vectors can improve reasoning ability even on tasks with graph structure that is *hidden*, rather than passed to the model in the form of edge vectors. We use the Sokoban (Guez et al., 2019) reinforcement learning task to investigate. In Sokoban, the agent must push four yellow boxes onto the red targets within 120 time steps. Humans solving these

puzzles tend to plan out which boxes will go onto which targets. Assuming that a successful agent will learn a similar strategy, representing each box-to-target pair as a directed relation, we hypothesize that RT is more capable than a standard transformer at reasoning over such pairwise relations. For evaluation, we use RT in place of the standard transformer originally used by the Working Memory Graph (WMG) RL agent (Loynd et al., 2020), then train both modified and unmodified agents on the Sokoban task for 10 million time steps.

**Main Results** We find that using RT reduces the agent's final error rate by a relative **15%**, from 34% to 29% of the puzzles. This improvement is much larger than the confidence intervals (0.3% and 0.7% standard error, respectively). Our results support the hypothesis that RT can learn even hidden graph structure. One alternative explanation would be that the extra trainable parameters added by RT simply improved the expressivity of WMG. But this seems unlikely since hyperparameter tuning of the unmodified WMG agent (Loynd et al., 2020) converged to an intermediate model size, rather than a larger model for more expressivity.

## C  HYPERPARAMETERS

To tune the hyperparameters of RT and the CLRS-30 baseline GNNs, we used Distributed Grid Descent (DGD) (Loynd et al., 2020), a self-guided form of random search. Each search was terminated after model performance converged to a stable value. Then 20 additional runs were executed, using the winning hyperparameter configuration, to obtain results free from selection bias. All tuning runs used the CLRS-30 protocol described in D, except for the following details:

1. To prevent tuning on the canonical datasets or any fixed datasets at all, the dataset generation seeds were randomized at the start of each run.

2. To reduce variance, the minimum evaluation dataset size was raised from 32 to 100.

3. To mitigate the computational costs, all models were tuned on only the 8 core algorithms, and each training run was shortened to 32,000 examples.

Very similar procedures were used to tune RT hyperparameters for the other (non-CLRS-30) experiments. For all experiments, all untuned hyperparameter values were chosen to match the settings of the corresponding baseline models.

### C.1  CLRS-30 EXPERIMENTS

Table 2 lists the tuned hyperparameter values for CLRS-30 experiments, and Table 3 reports the sets of values considered in those searches. The runtime sizes of the corresponding models are found in Table 4, along with their training speeds in Table 5.

Table 2: Tuned hyperparameter values for CLRS-30 experiments.

|  | Deep Sets | GAT-v1 | GAT-v2 | MPNN | PGN-u | PGN-m | RT |
|---|---|---|---|---|---|---|---|
| batch_size | 4 | 4 | 4 | 4 | 4 | 4 | 4 |
| num_layers | 2 | 2 | 2 | 1 | 1 | 3 | 3 |
| learning_rate | 2.5e-4 | 6.3e-4 | 2.5e-4 | 1.6e-3 | 1.6e-3 | 1e-3 | 2.5e-4 |
| $d_n = d_e = d_g$ | 512 | - | - | 512 | 180 | 512 | - |
| nb_heads | - | 10 | 12 | - | - | - | 12 |
| head_size | - | 64 | 64 | - | - | - | 16 |
| $d_{nh}$ | - | - | - | - | - | - | 32 |
| $d_{eh1}$ | - | - | - | - | - | - | 16 |
| $d_{eh2}$ | - | - | - | - | - | - | 8 |
| ptr_from_edges | - | - | - | - | - | - | true |
| graph_vec | - | - | - | - | - | - | cat |

Table 3: Hyperparameter values considered for CLRS-30 experiments.

| | |
|---|---|
| batch_size | 1, 2, 4, 8, 16 |
| num_layers | 1, 2, 3, 4 |
| learning_rate | 4e-5, 6.3e-5, 1e-4, 1.6e-4, 2.5e-4, 4e-4, 6.3e-4, 1e-3, 1.6e-3, 2.5e-3, 4e-3, 6.3e-3, 1e-2 |
| $d_n = d_e = d_g$ | 45, 64, 90, 128, 180, 256, 360, 512 |
| nb_heads | 3, 4, 6, 8, 10, 12, 16 |
| head_size | 8, 12, 16, 24, 32, 45, 64 |
| $d_{nh}$ | 4, 6, 8, 12, 16, 24, 32, 45, 64, 90 |
| $d_{eh1}$ | 12, 16, 24, 32, 45, 64 |
| $d_{eh2}$ | 4, 6, 8, 12, 16, 24, 32, 45 |
| ptr_from_edges | false, true |
| graph_vec | core, cat |

Table 4: Number of trainable parameters in each model tested on CLRS-30, including the framework's encoding and decoding layers, on the reference algorithm Bellman Ford.

| | Deep Sets | GAT-v1 | GAT-v2 | MPNN | PGN-u | PGN-m | RT |
|---|---|---|---|---|---|---|---|
| tuned | 8,675,332 | 11,168,732 | 17,730,076 | 6,573,060 | 816,844 | 10,777,604 | 1,103,404 |
| untuned | 414,468 | 418,960 | 397,957 | 414,468 | 414,468 | 414,468 | - |

Table 5: Training speed in examples per second on a T4 GPU, on the reference algorithm Bellman Ford.

| | Deep Sets | GAT-v1 | GAT-v2 | MPNN | PGN-u | PGN-m | RT |
|---|---|---|---|---|---|---|---|
| tuned | 13.4 | 14.8 | 10.1 | 15.7 | 18.7 | 11.6 | 8.7 |
| untuned | 128.9 | 140.7 | 134.8 | 128.8 | 128.6 | 128.6 | - |

## C.2 LOBSTER GRAPH HYPERPARAMETERS

Table 6 lists the tuned hyperparameter values for RT on the shortest-path task over lobster graphs, along with the sets of values considered in that search.

## C.3 SOKOBAN HYPERPARAMETERS

Table 7 lists the tuned hyperparameter values for RT on the Sokoban task, along with the sets of values considered in that search.

## D TRAIN/TEST PROTOCOL

Except where specifically noted, our experiments follow the exact train/test protocol defined by CLRS-30. CLRS-30 provides canonical datasets (training, validation, and test) which can also be generated from specific random seeds: 1, 2, 3. The graphs in the training and validation datasets contain 16 nodes, while the test graphs are of size 64 to evaluate the out-of-distribution (OOD) generalization of models. Training is performed on a random sequence of 320,000 examples drawn with replacement from the train set, where each example trajectory contains a variable number of reasoning steps. During training, the model is evaluated on the validation set after every 320 examples. At the end of training, the model with the highest validation score is evaluated on the test set. The average test micro-F1 score is reported for all results. The published CLRS-30 results are reported as averages over 3 random run seeds, but we use 20 seeds in all of our experiments.

## E TEST RESULTS ON CLRS-30

Test performance of all tuned models evaluated on CLRS-30 may be found in Tables 19 (mean test micro-F1 score) and 18 (standard deviation). On certain tasks in which baseline GNN performances

Table 6: Tuned hyperparameter values for RT on the lobster graph experiment.

|  | Tuned values | Values considered |
|---|---|---|
| learning_rate | 6.3e-5 | 4e-5, 6.3e-5, 1e-4, 1.6e-4, 2.5e-4, 4e-4 |
| output_layer_size | 180 | 64, 90, 128, 180, 256, 360, 512 |
| nb_heads | 8 | 3, 4, 6, 8, 10 |
| head_size | 28 | 12, 16, 20, 24, 28, 32 |
| $d_{nh}$ | 12 | 3, 4, 6, 8, 12, 16, 24 |
| $d_e$ | 128 | 16, 24, 32, 45, 64, 90, 128 |
| $d_{eh1}$ | 32 | 12, 16, 24, 32, 45, 64, 90 |
| $d_{eh2}$ | 8 | 8, 12, 16, 24, 32, 45, 64 |
| dropout_rate | 0.0 | 0.0, 0.01, 0.02, 0.04, 0.08 |
| grad_clip | 128.0 | 16.0, 32.0, 64.0, 128.0, 256.0, 512.0 |

Table 7: Tuned hyperparameter values for RT on Sokoban.

|  | Tuned values | Values considered |
|---|---|---|
| $d_e$ | 128 | 8, 12, 16, 24, 32, 45, 64, 90, 128, 180, 256, 360, 512, 720, 1024 |
| $d_{eh1}$ | 64 | 6, 8, 12, 16, 24, 32, 45, 64, 90, 128, 180, 256, 360, 512 |
| $d_{eh2}$ | 12 | 6, 8, 12, 16, 24, 32, 45, 64, 90, 128, 180, 256, 360, 512 |

improved, we also observed higher variability in results. Following Veličković et al. (2022), we report the best performance between PGN-u and PGN-m on every task in the column titled PGN-c (for PGN-combination). Note therefore that PGN-c does not represent a single model.

We bold the results of the best-performing single model for each specific task. RT was the best-performing on **11 out of 30** of the tasks. Overall, MPNN was the top-scoring baseline model, winning on 8 tasks. Deep Sets won on 4 tasks. PGN-u won on 1 task, and PGN-m won on 5 tasks. Finally, GAT-v2 won on 1 task.

Table 1 shows test performances of all tuned models across 8 algorithm classes. As mentioned in Section 5.1.3, RT performed best in **6 out of 8** of the classes, and second-best on greedy algorithms and sorting.

Table 8 shows which algorithms belong to which classes, and which task types. Each algorithm may correspond to multiple task types if the hint and final outputs of that algorithm differ.

Table 9 shows test performance of the baseline GNN models trained on datasets of varying size, without hyperparameter tuning. Specifically, we compare test scores between the models trained on datasets of size 1000, and datasets of size 10000. We note that the results in the first row agree closely with those in the CLRS-30 paper (Table 11).

## F   CLRS-30 BENCHMARK ABLATIONS

We perform our ablation studies on the 8 core CLRS-30 tasks. Table 13 compares the test performance of the standard transformer to the performance of RT. Table 14 shows the drop in test performance that resulted from restricting RT to one layer. The original, tuned RT had 3 layers, and is labeled RT-3. Table 15 shows the marginal improvement that resulted from decoding node vectors using only edge vectors. Table 16 shows the effects that handling global input vectors through a core node (vs. concatenation with the input node vectors) has on RT test performance. RT with core node only won on 3 out of 7 tasks, but had the higher average test performance by 0.08%. Finally, Table 17 shows the drop in test performance that resulted from removing edge updates from RT.

## G   CLRS-30 MODEL VARIANCE

Table 18 shows that RT produces some of the largest standard deviations over random runs. To investigate these results, we first examine the Naïve String Matcher task, where RT's variance is the largest value in the table (32.3%). Figure 5 displays each model's distribution of run scores, through

histograms. We see that RT's twenty runs on this task obtain a wide range of scores, including a spike in the range from 80% to 100%, while none of the baseline model runs surpass a score of 20%. This explains RT's high variance on this task compared to the other models. In summary, in a case like this higher variance is a consequence of higher performance.

The same behavior can be inferred without the histograms by examining Table 18 and Table 19, which show that RT's high variance on Naïve String Matcher is associated with far higher mean performance on the task than the baseline models. The same pattern is apparent for RT's next two highest-variance tasks, Topological Sort and Quickselect.

For a broader view of variance, Figure 6 displays the model score distributions over all CLRS-30 tasks. We see that RT's distribution is weighted more heavily than the other models in the >80% range, and underweighted in the <20% range. Meanwhile, all model distributions are spread quite widely over the five range bins. This spread is quantified numerically by the last row of Table 18, where RT's overall standard deviation is shown to be one of the smallest.

Figure 5: Histograms describing the variance in model results on Naïve String Matcher, across runs

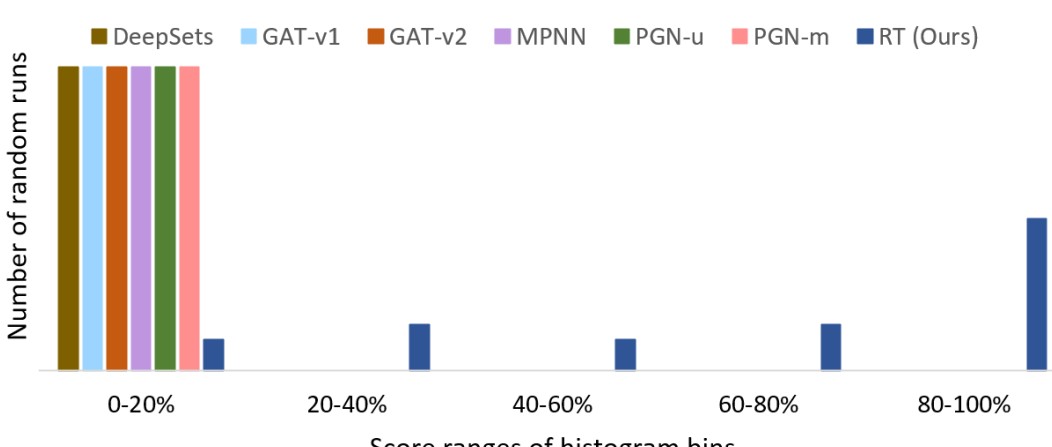

Figure 6: Histograms describing the variance in model results across all runs on the 30 CLRS tasks

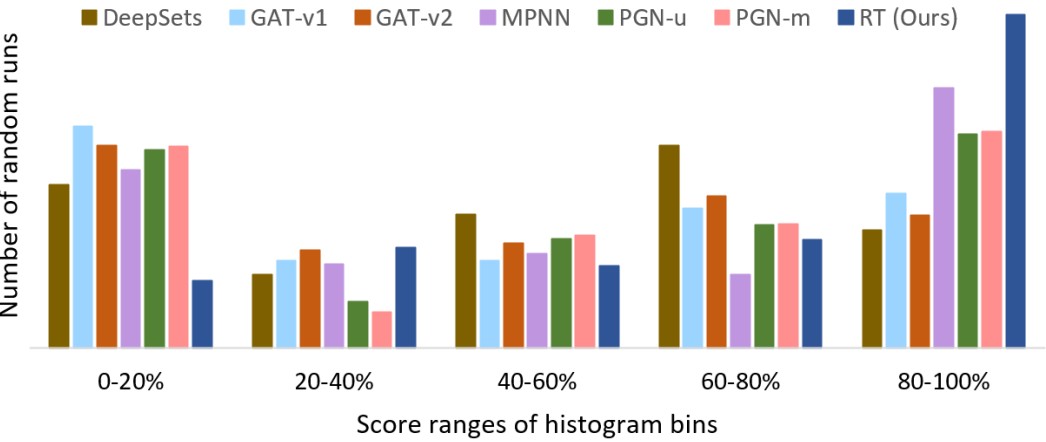

Table 8: Algorithms and their respective classes and task types. Taking hint targets into consideration, there are 3 task types in CLRS-30: node (N), edge (E), and graph (G).

| Algorithm | N | E | G | D&C | DP | Geo | Graph | Greedy | Search | Sort | String |
|---|---|---|---|---|---|---|---|---|---|---|---|
| Activity Selector | ■ | ■ | | | | | | ■ | | | |
| Articulation Points | ■ | ■ | ■ | | | | ■ | | | | |
| Bellman-Ford | ■ | ■ | | | | | ■ | | | | |
| BFS | ■ | ■ | | | | | ■ | | | | |
| Binary Search | ■ | ■ | | | | | | | ■ | | |
| Bridges | ■ | ■ | ■ | | | | ■ | | | | |
| Bubble Sort | ■ | ■ | | | | | | | | ■ | |
| DAG Shortest Paths | ■ | ■ | ■ | | | | ■ | | | | |
| DFS | ■ | ■ | ■ | | | | ■ | | | | |
| Dijkstra | ■ | ■ | | | | | ■ | | | | |
| Find Max. Subarray | ■ | ■ | ■ | ■ | | | | | | | |
| Floyd-Warshall | ■ | ■ | | | | | ■ | | | | |
| Graham Scan | ■ | ■ | ■ | | | ■ | | | | | |
| Heapsort | ■ | ■ | | | | | | | | ■ | |
| Insertion Sort | ■ | ■ | | | | | | | | ■ | |
| Jarvis' March | ■ | ■ | | | | ■ | | | | | |
| KMP Matcher | ■ | ■ | | | | | | | | | ■ |
| LCS Length | | ■ | | | ■ | | | | | | |
| Matrix Chain Order | | ■ | | | ■ | | | | | | |
| Minimum | ■ | ■ | | | | | | | ■ | | |
| MST-Kruskal | ■ | ■ | ■ | | | | ■ | | | | |
| MST-Prim | ■ | ■ | | | | | ■ | | | | |
| Naïve String Match | ■ | ■ | | | | | | | | | ■ |
| Optimal BST | | ■ | | | ■ | | | | | | |
| Quickselect | ■ | ■ | ■ | | | | | | ■ | | |
| Quicksort | ■ | ■ | | | | | | | | ■ | |
| Segments Intersect | ■ | | ■ | | | ■ | | | | | |
| SCC | ■ | ■ | ■ | | | | ■ | | | | |
| Task Scheduling | ■ | ■ | ■ | | | | | ■ | | | |
| Topological Sort | ■ | ■ | ■ | | | | ■ | | | | |
| Total | 27 | 29 | 14 | 1 | 3 | 3 | 12 | 2 | 3 | 4 | 2 |

Table 9: Average test scores of untuned baseline models trained on either the original small datasets, versus trained on the expanded training datasets, for the 8 core tasks.

| Training Set Size | Deep Sets | GAT-v1 | GAT-v2 | MPNN | PGN-u | PGN-m | PGN-c |
|---|---|---|---|---|---|---|---|
| 1000 | 45.95% | 42.24% | 42.53% | 54.66% | 51.07% | 56.18% | 61.29% |
| 10000 | 46.70% | 42.97% | 45.37% | 56.93% | 56.77% | 58.20% | 67.11% |

Table 10: Test score improvements from hyperparameter tuning on the 8 core tasks. The drop in score for PGN-m was likely the result of tuning hyperparameters (for all models) on shorter training runs than was used for evaluation.

| Algorithm | Deep Sets | GAT-v1 | GAT-v2 | MPNN | PGN-u | PGN-m | PGN-c |
|---|---|---|---|---|---|---|---|
| No Tuning | 46.70% | 42.97% | 45.37% | 56.93% | 56.77% | 58.20% | 67.11% |
| Tuning | 51.55% | 50.47% | 48.97% | 59.75% | 59.97% | 57.53% | 67.86% |

Table 11: Reproduction of baseline model results on the 8 core algorithms.

| Results | # Seeds | Deep Sets | MPNN | PGN-c |
|---|---|---|---|---|
| Published (Veličković et al., 2022) | 3 | 45.50% | 53.05% | 61.15% |
| Reproduced (Ours) | 20 | 45.95% | 54.66% | 61.29% |

Table 12: Test score improvements from augmenting the CLRS-30 baseline models.

| Results | # Seeds | Deep Sets | MPNN | PGN-c |
|---|---|---|---|---|
| CLRS-30 Published Results | 3 | 42.72% | 44.99% | 50.84% |
| Our Results | 20 | 48.60% | 49.23% | 52.82% |

Table 13: Standard transformer ablation (with re-tuned hyperparameters) evaluated on the 8 core algorithms.

| Algorithm | Transformer | RT (Ours) |
|---|---|---|
| Activity Selector | $68.01\% \pm 6.0$ | $\mathbf{87.72\%} \pm 2.7$ |
| Bellman-Ford | $39.79\% \pm 2.4$ | $\mathbf{94.24\%} \pm 1.5$ |
| Binary Search | $2.84\% \pm 0.6$ | $\mathbf{81.48\%} \pm 6.7$ |
| Find Max. Subarray | $26.60\% \pm 6.0$ | $\mathbf{66.52\%} \pm 3.7$ |
| Graham Scan | $58.84\% \pm 6.9$ | $\mathbf{74.15\%} \pm 7.4$ |
| Insertion Sort | $60.57\% \pm 15.7$ | $\mathbf{89.43\%} \pm 9.0$ |
| Matrix Chain Order | $80.54\% \pm 6.8$ | $\mathbf{91.89\%} \pm 1.2$ |
| Naïve String Match | $1.51\% \pm 1.7$ | $\mathbf{65.01\%} \pm 32.$ |
| Average | 42.34% | **81.30%** |

Table 14: Single-layer RT ablation (with re-tuned hyperparameters) evaluated on the 8 core algorithms.

| Algorithm | RT-1 | RT-3 (Ours) |
|---|---|---|
| Activity Selector | $83.41\% \pm 5.8$ | $\mathbf{87.72\%} \pm 2.7$ |
| Bellman-Ford | $93.52\% \pm 1.7$ | $\mathbf{94.24\%} \pm 1.5$ |
| Binary Search | $61.61\% \pm 15.1$ | $\mathbf{81.48\%} \pm 6.7$ |
| Find Max. Subarray | $65.07\% \pm 4.0$ | $\mathbf{66.52\%} \pm 3.7$ |
| Graham Scan | $\mathbf{74.80\%} \pm 7.9$ | $74.15\% \pm 7.4$ |
| Insertion Sort | $77.54\% \pm 10.3$ | $\mathbf{89.43\%} \pm 9.0$ |
| Matrix Chain Order | $91.05\% \pm 1.0$ | $\mathbf{91.89\%} \pm 1.2$ |
| Naïve String Match | $16.50\% \pm 23.2$ | $\mathbf{65.01\%} \pm 32.$ |
| Average | 70.44% | **81.30%** |

Table 15: Ablation of the node pointer decoding procedure evaluated on the 8 core algorithms.

| Algorithm | RT with original decoding | RT with edge-only decoding (Ours) |
|---|---|---|
| Activity Selector | $\mathbf{88.09\%} \pm 5.0$ | $87.72\% \pm 2.7$ |
| Bellman-Ford | $\mathbf{94.95\%} \pm 1.1$ | $94.24\% \pm 1.5$ |
| Binary Search | $80.06\% \pm 6.6$ | $\mathbf{81.48\%} \pm 6.7$ |
| Find Max. Subarray | $\mathbf{67.44\%} \pm 3.7$ | $66.52\% \pm 3.7$ |
| Graham Scan | $\mathbf{75.71\%} \pm 10.3$ | $74.15\% \pm 7.4$ |
| Insertion Sort | $71.32\% \pm 11.0$ | $\mathbf{89.43\%} \pm 9.0$ |
| Matrix Chain Order | $\mathbf{92.12\%} \pm 0.9$ | $91.89\% \pm 1.2$ |
| Naïve String Match | $\mathbf{78.43\%} \pm 22.5$ | $65.01\% \pm 32.$ |
| Average | 81.00% | **81.30%** |

Table 16: Ablation of RT core node (vs. concatenation of the global vector) evaluated on 7 representative algorithms that use global input features or hints.

| Algorithm | RT with core node | RT without core node (Ours) |
|---|---|---|
| Binary Search | $75.40\% \pm 11.0$ | $\mathbf{81.48\%} \pm 6.7$ |
| Find Max. Subarray | $\mathbf{66.96\%} \pm 4.5$ | $66.52\% \pm 3.7$ |
| Graham Scan | $71.83\% \pm 10.2$ | $\mathbf{74.15\%} \pm 7.4$ |
| Heapsort | $30.67\% \pm 18.9$ | $\mathbf{32.96\%} \pm 14.8$ |
| KMP Matcher | $0.02\% \pm 0.0$ | $\mathbf{0.03\%} \pm 0.1$ |
| MST-Kruskal | $\mathbf{75.59\%} \pm 5.8$ | $64.91\% \pm 11.8$ |
| Task Scheduling | $\mathbf{83.09\%} \pm 1.8$ | $82.93\% \pm 1.8$ |
| Average | **57.65%** | 57.57% |

Table 17: Ablation of RT's edge update procedure evaluated on the 8 core algorithms

| Algorithm | RT without edge updates | RT with edge updates (Ours) |
|---|---|---|
| Activity Selector | 82.13% ± 7.0 | **87.72%** ± 2.7 |
| Bellman-Ford | 91.03% ± 2.3 | **94.24%** ± 1.5 |
| Binary Search | 60.29% ± 12.5 | **81.48%** ± 6.7 |
| Find Max. Subarray | 19.09% ± 1.6 | **66.52%** ± 3.7 |
| Graham Scan | 69.19% ± 7.7 | **74.15%** ± 7.4 |
| Insertion Sort | 22.30% ± 8.5 | **89.43%** ± 9.0 |
| Matrix Chain Order | 85.68% ± 1.5 | **91.89%** ± 1.2 |
| Naïve String Match | 2.22% ± 0.8 | **65.01%** ± 32. |
| Average | 53.99% | **81.30%** |

Table 18: Standard deviations over 20 seeds for all tuned models on all algorithms. Each value in the row "Over All Runs" is not an average of variances for each algorithm, but rather the variance across all runs.

| Algorithm | Deep Sets | GAT-v1 | GAT-v2 | MPNN | PGN-u | PGN-m | RT (Ours) |
|---|---|---|---|---|---|---|---|
| Activity Selector | 1.7% | 1.4% | 2.1% | 1.3% | 2.6% | 1.9% | 2.7% |
| Articulation Points | 6.0% | 8.0% | 6.3% | 6.1% | 3.5% | 7.5% | 14.6% |
| Bellman-Ford | 2.4% | 1.5% | 1.6% | 1.9% | 0.9% | 1.3% | 1.5% |
| BFS | 1.0% | 0.4% | 0.4% | 0.2% | 0.3% | 0.5% | 0.7% |
| Binary Search | 3.8% | 5.8% | 7.9% | 5.0% | 10.4% | 3.7% | 6.7% |
| Bridges | 4.8% | 6.4% | 8.0% | 17.8% | 11.0% | 7.8% | 11.8% |
| Bubble Sort | 3.1% | 2.5% | 1.3% | 5.0% | 1.9% | 0.2% | 13.0% |
| DAG Shortest Paths | 3.8% | 2.3% | 2.2% | 1.6% | 3.2% | 0.9% | 1.6% |
| DFS | 1.9% | 2.1% | 1.7% | 2.7% | 1.5% | 1.5% | 10.5% |
| Dijkstra | 3.6% | 9.5% | 7.1% | 4.3% | 11.0% | 3.4% | 5.8% |
| Find Max. Subarray | 1.5% | 2.4% | 2.1% | 2.9% | 1.6% | 15.0% | 3.7% |
| Floyd-Warshall | 4.2% | 6.0% | 3.7% | 4.7% | 4.8% | 2.5% | 7.6% |
| Graham Scan | 3.3% | 6.6% | 5.3% | 1.4% | 5.7% | 6.0% | 7.4% |
| Heapsort | 8.8% | 3.8% | 4.1% | 7.1% | 8.9% | 0.4% | 14.8% |
| Insertion Sort | 1.5% | 14.8% | 11.6% | 9.1% | 11.2% | 0.2% | 9.0% |
| Jarvis' March | 8.6% | 2.6% | 3.9% | 29.3% | 2.5% | 5.8% | 2.2% |
| KMP Matcher | 1.0% | 0.6% | 0.6% | 1.1% | 0.7% | 0.4% | 0.1% |
| LCS Length | 6.5% | 7.2% | 4.6% | 2.7% | 7.8% | 3.5% | 4.1% |
| Matrix Chain Order | 2.8% | 5.4% | 5.1% | 2.3% | 0.9% | 1.1% | 1.2% |
| Minimum | 5.8% | 2.6% | 10.4% | 3.4% | 3.4% | 27.3% | 2.0% |
| MST-Kruskal | 4.9% | 2.9% | 4.2% | 5.9% | 6.1% | 4.3% | 11.8% |
| MST-Prim | 11.4% | 9.2% | 10.7% | 5.1% | 4.6% | 7.2% | 7.9% |
| Naïve String Match | 0.3% | 0.7% | 0.5% | 1.7% | 0.8% | 0.7% | 32.3% |
| Optimal BST | 6.0% | 5.7% | 5.1% | 3.5% | 2.0% | 2.0% | 2.6% |
| Quickselect | 2.4% | 0.5% | 0.4% | 1.6% | 0.9% | 1.0% | 17.3% |
| Quicksort | 4.3% | 1.3% | 0.8% | 3.9% | 1.9% | 0.2% | 13.2% |
| Segments Intersect | 0.7% | 0.5% | 0.7% | 2.5% | 0.8% | 0.8% | 2.6% |
| SCC | 5.9% | 8.8% | 9.9% | 5.2% | 6.0% | 6.9% | 15.2% |
| Task Scheduling | 0.5% | 0.4% | 0.8% | 0.9% | 0.5% | 0.5% | 1.8% |
| Topological Sort | 10.3% | 12.7% | 17.8% | 5.8% | 6.5% | 10.1% | 17.5% |
| Over All Runs | 29.3% | 32.0% | 31.3% | 34.6% | 33.1% | 35.0% | 29.6% |

Table 19: Average test scores of all tuned models on all algorithms.

| Algorithm | Deep Sets | GAT-v1 | GAT-v2 | MPNN | PGN-u | PGN-m | PGN-c | RT (Ours) |
|---|---|---|---|---|---|---|---|---|
| Activity Selector | 72.22% | 68.89% | 67.45% | **95.45%** | 67.05% | 69.83% | 69.83% | 87.72% |
| Articulation Points | 38.50% | 31.46% | 31.96% | 46.21% | 46.87% | **49.73%** | 49.73% | 34.15% |
| Bellman-Ford | 51.00% | 93.10% | 93.75% | 95.42% | **95.83%** | 95.43% | 95.83% | 94.24% |
| BFS | 98.14% | 99.76% | 99.49% | **99.78%** | 99.71% | 99.27% | 99.71% | 99.14% |
| Binary Search | 55.77% | 17.21% | 31.11% | 38.00% | 61.71% | **87.03%** | 87.03% | 81.48% |
| Bridges | 36.20% | 22.90% | 24.07% | **61.29%** | 44.72% | 57.15% | 57.15% | 37.88% |
| Bubble Sort | **65.03%** | 9.69% | 8.46% | 13.14% | 7.54% | 2.05% | 7.54% | 38.22% |
| DAG Shortest Paths | 77.98% | 86.44% | 89.35% | 97.33% | 96.23% | **98.16%** | 98.16% | 96.61% |
| DFS | 7.62% | 11.71% | 12.08% | 13.85% | 10.78% | 8.88% | 10.78% | **39.23%** |
| Dijkstra | 44.51% | 61.42% | 68.10% | **92.18%** | 78.45% | 90.02% | 90.02% | 91.20% |
| Find Max. Subarray | 12.29% | 15.19% | 14.80% | 16.14% | 16.71% | 51.30% | 51.30% | **66.52%** |
| Floyd-Warshall | 6.24% | 30.53% | 37.58% | 28.81% | 27.60% | **39.24%** | 39.24% | 31.59% |
| Graham Scan | 64.48% | 65.05% | 67.07% | **95.31%** | 67.89% | 64.66% | 67.89% | 74.15% |
| Heapsort | **72.39%** | 9.90% | 7.65% | 27.08% | 17.88% | 1.60% | 17.88% | 32.96% |
| Insertion Sort | 72.88% | 61.00% | 50.19% | 50.30% | 82.42% | 3.64% | 82.42% | **89.43%** |
| Jarvis' March | 46.01% | 55.40% | 50.23% | 59.31% | 49.54% | 49.57% | 49.57% | **94.57%** |
| KMP Matcher | **3.71%** | 1.03% | 1.32% | 2.35% | 1.75% | 0.44% | 1.75% | 0.03% |
| LCS Length | 55.85% | 48.14% | 48.30% | 54.42% | 53.85% | 56.00% | 56.00% | **83.32%** |
| Matrix Chain Order | 81.65% | 81.59% | 65.58% | 85.53% | 86.65% | 86.47% | 86.65% | **91.89%** |
| Minimum | 91.69% | 86.87% | 80.60% | 90.73% | 89.22% | 70.69% | 89.22% | **95.28%** |
| MST-Kruskal | 69.07% | 67.26% | 68.60% | **70.84%** | 63.47% | 69.35% | 69.35% | 64.91% |
| MST-Prim | 35.70% | 61.83% | 68.72% | **86.43%** | 81.59% | 83.61% | 83.61% | 85.77% |
| Naïve String Match | 2.12% | 1.70% | 1.81% | 1.83% | 1.46% | 1.90% | 1.90% | **65.01%** |
| Optimal BST | 67.37% | 61.91% | 63.78% | 66.48% | 65.19% | 70.74% | 70.74% | **74.40%** |
| Quickselect | 5.51% | 2.03% | 2.40% | 3.07% | 2.00% | 4.92% | 4.92% | **19.18%** |
| Quicksort | **65.26%** | 4.41% | 5.00% | 17.95% | 7.90% | 2.63% | 7.90% | 39.42% |
| Segments Intersect | 85.94% | 86.36% | 86.10% | **94.46%** | 85.89% | 85.66% | 85.89% | 84.94% |
| SCC | 21.13% | 23.66% | **29.47%** | 12.58% | 12.97% | 17.20% | 17.20% | 28.59% |
| Task Scheduling | 83.45% | 82.61% | 82.61% | 83.36% | 83.54% | **83.62%** | 83.62% | 82.93% |
| Topological Sort | 19.02% | 43.47% | 43.23% | 54.93% | 51.71% | 64.28% | 64.28% | **80.62%** |
| Average | 50.29% | 46.42% | 46.70% | 55.15% | 51.94% | 52.17% | 56.57% | **66.18%** |

