# OpenReview forum: "Relational Attention: Generalizing Transformers for Graph-Structured Tasks"
_ICLR.cc/2023/Conference — ICLR 2023 notable top 25%_

### Official Review · Reviewer_H4RY · 2022-10-16

**Confidence:** 4
**Correctness:** 4
**Technical Novelty And Significance:** 3
**Empirical Novelty And Significance:** 3
**Recommendation:** 8

**Clarity, Quality, Novelty And Reproducibility:**

### Clarity
This paper does a good job of clearly explaining their Relational Transformer model and its relationship with graph neural networks. It also clearly describes the experiments and their results.

A few minor clarity/editing issues I noticed:
- In section 3.2: I don't know what the authors mean by "Since these four input vectors are not interchangeable, their message projections can be aggregated through summation instead of attention", since projections can be aggregated through summation even when the vectors are interchangeable. Also, the authors seem to actually aggregate them via concatenation followed by matmul (e.g. via a learned linear combination), not via ordinary summation.
- In section 4, the "graphormer" architecture ([Ying et al. 2021](https://arxiv.org/abs/2106.05234)) is referred to as "graphformer", which is actually the name of a different GNN-transformer hybrid ([Yang et al. 2021](https://arxiv.org/abs/2105.02605)).
- In section 5.1.3 there is a missing figure reference (Figure ??)

### Quality
The proposed architecture seems very reasonable and well motivated, without unnecessary heuristics, which makes it more likely that this architecture will be used by future work. I like how the edge update equation is similar to a transformer MLP block, but does not use full attention so that the complexity stays O(n^2) with respect to the number of nodes.

The experiments also appear to be well done and quite thorough. The authors conduct a large hyperparameter search with multiple ablations, and give additional information (including standard deviations) in the appendix.

### Novelty

The proposed architecture clearly takes a lot of inspiration from both the standard transformer architecture and the "graph networks" framework for GNN models (an intentional choice by the authors). The main differences from prior work come in how they handle edge vectors. I see three main differences:
1. The RT architecture uses full learned weight matrices to transform relational features into attention weights and attention values, whereas previous work on relative attention (e.g. Shaw et al. (2018) and Dai et al. (2019)) often directly uses relational features to bias attention.
2. The relational features are explicitly intended to correspond to correspond to edges, whereas Shaw et al. (2018) and Dai et al. (2019) primarily use them to encode relative node position information. (One related work in this space that the authors do not cite is [Hellendoorn et al. (2019)](https://openreview.net/pdf?id=B1lnbRNtwr) which adapts the mechanism of Shaw et al. to correspond to edges in a graph similar to how it is done here.)
3. The edge features can be updated at each step, using the features of incident nodes and reverse-directed edges. This gives the model more flexibility than earlier relational attention models, without changing the O(n^2) complexity.

I think the TokenGT model of Kim et al. (2022) is another interesting point of comparison for this work, as both the RT and TokenGT models seek to make minimal changes to a transformer to apply them to graph tasks. The RT model proposed here does this by treating nodes like tokens and modifying the update equations to explicitly use and edit edge features, whereas the TokenGT model treats both nodes and edges as tokens and uses random vectors (or spectral embeddings) to communicate edge relationships. Both approaches seem reasonable to me, and it's interesting to see a different perspective on how to apply transformers to graph tasks in a generic way.

Empirically, applying a transformer-based model to algorithmic reasoning tasks such as the CLRS benchmark is novel as far as I am aware, and the experimental results provide good new evidence that transformer-based graph models are well suited to this kind of task. (However, the experimental results do not compare between different transformer-based models, so it's not clear how good the RT architecture is within the broader family of transformer-based graph models, e.g. relative to TokenGT, Graphormer, or any of the other related works mentioned in the paper.)

### Reproducibility
The architecture is explained well, so I think it would be possible to reimplement it successfully. The authors also plan to release the implementation itself open-source.

**Strength And Weaknesses:**

Strengths
- The paper is well written and easy to follow.
- The proposed architecture is quite straightforward and generic, seeming to combine the most important parts of a standard transformer and of the "graph networks" framework without adding complex heuristic adjustments.
- The experiments are very thorough, and the authors describe a number of adjustments they made to improve the performance of the baselines, which makes the strong performance of their proposed model more impressive.

Weakesses
- **(addressed by new revision)** The framing of the paper in the abstract and introduction does not do enough to contextualize this work relative to the extensive previous work on incorporating relational attention into transformers, in my opinion.
    - The authors claim that transformers are at a disadvantage for graphs, that they have "not had the same success" as GNNs in graph-structured domains, that there is "no generally applicable way of passing edge feature vectors into a transformer without losing the graph's connectivity information", and that their work addresses these shortcomings. However, there have been a number of recent works that demonstrate that transformers succeed in graph structured domains, and multiple proposals for passing edge feature vectors into transformers. (The authors discuss some of this in the related work section, which is good. But I think the claims in the introduction should be substantially reduced based on the prior work in this area.)
- **(mostly addressed by new revision)** Although the authors discuss differences between their model and previous proposals for adding relational attention to transformers (calling most of them "partially relational"), they do not include any experiments comparing against these models. It's not obvious to me that the distinction between partially relational and fully relational models is meaningful, or that their approach would work better.
    - Adding additional experiments comparing against some subset of partially-relational transformers would strengthen the paper. But I would also be satisfied if the authors add to the paper a justification for not including these experiments. (For instance, perhaps the authors only want to claim that transformer variants can work better than GNNs on CLRS and don't want to claim that their model is superior to the "partially relational" versions? Or perhaps there is some reason why previous works are not well suited for this task?)
    - I'd be particularly interested in seeing how the RT models proposed here fare against the recently-proposed TokenGT architecture, since that architecture is a different way of generalizing transformers to operate over graphs in a general-purpose way.
    - *The revised version of this paper makes it clear that the authors do not intend to claim that their model outperforms other graph-transformer variants in general, and that the code for other graph-transformer models can't be directly applied to the CLRS benchmark used here, which requires JAX code. I still think it would be very interesting to see how this model compares to other approaches like TokenGT, and that those results would improve the paper, but I don't think they are necessary to include.*

**Summary Of The Paper:**

This paper introduces a new variant of transformers for graph-structured tasks, building on both the original transformer architecture and the "graph networks" framework of [Battaglia et al. (2021)](https://arxiv.org/abs/1806.01261). Their model, called "relational transformer" (RT), includes edge attributes that participate in the attention mechanism and that are updated at each layer similarly to the transformer's MLP block.

The authors compare their architecture to the standard set of baseline GNN models for the CLRS algorithmic reasoning benchmark task, and also modify the benchmark to make its interface more expressive. The experiments appear to be very thorough, and the authors both increase the size of the training dataset and tune the hyperparameters for baseline models to ensure that the results accurately reflect the ability of the models to generalize to out-of-distribution graph problems. The authors also include several ablations of their model. Overall, the experiments show that their RT model outperforms the GNN baselines on many of the tasks by a large margin (but not all), and that the edge update mechanism has a significant impact.

The authors also evaluate their model's ability to solve an end-to-end shortest path problem (where step-by-step supervision is not provided) and a Sokoban task (where no graph structure is provided and the model must learn it on its own). On both of these tasks, the authors again find improvements from incorporating their model.

**Summary Of The Review:**

*Initial submission:*
The architecture proposed here is interesting and well motivated, and the experimental results are thorough and demonstrate that the model works well relative to ordinary GNNs, which are both reasons for acceptance. My main reservation (and why I gave it a 6 instead of an 8) is that the paper does not currently engage much with prior work on transformer-based graph models (such as TokenGT and Graphormer) in either the introduction or the experiments section.

I think that the authors should acknowledge (in the introduction) what existing transformer graph models have already accomplished and discuss more clearly what their proposal provides beyond these models, instead of framing their ideas as if they are making the first attempt to test a totally new conjecture. The paper would also be stronger if it included empirical comparisons to some of these previous transformer graph models (or at least explained why those models aren't appropriate for the setting considered).

*Revision (as of Nov 12, 2022):*
The authors have substantially revised the introduction to better contextualize their contributions relative to prior work, and removed unsupported claims about how their model compares empirically with these other graph-transformer hybrids. I have thus raised my score to 8.

---

### Official Review · Reviewer_WhvK · 2022-10-22

**Confidence:** 3
**Correctness:** 4
**Technical Novelty And Significance:** 3
**Empirical Novelty And Significance:** 3
**Recommendation:** 8

**Clarity, Quality, Novelty And Reproducibility:**

The presentation quality is very good, the authors make their line of arguments easy to follow and all design decisions are clearly motivated.
The proposed changes are novel and reasonable. They directly model the property of graphs (nodes connected by edges) into the attention mechanism.
Regarding reproducibility, the authors provide the hyperparameters (and their range) for all experiments. They number the parameters of all models used in the experimental section and show all experimental results. The source code of the proposed architecture will be made publicly available. These points lead to good reproducibility of the proposed work.


**Strength And Weaknesses:**

Strengths:
The main strength of the presented work is the combination of the clear presentation and the well grounded motivation for the proposed approach. Furthermore, the extensive experimental section clearly shows that the proposed relational attention mechanism accounts for the improved performance in the benchmark tasks.

Weaknesses:
There are some minor formatting issues in the paper as of now.
On page 3, below equation 3, there seems to be a missing sentence or paragraph in front of the following section:
[“… where Ni denotes the set of …”]
There is also a missing figure (or reference to a figure) on page 7, section 5.1.3:
[“… when results are averaged over those classes (Figure ??).”


**Summary Of The Paper:**

The authors propose an adaption of transformer attention to graph input. Normally, a transformer takes an unordered set as input, which can not adequately represent the structured nature of a graph. The key change they propose is the inclusion of the directed edge between nodes into the attention mechanism.
The authors show the effectiveness of their proposed architecture through extensive experiments on the CLRS benchmark. Across most tasks, the proposed relational Transformer achieves competitive or better performance than the SOTA approaches.
An ablation study proves that the performance improvements are caused by the proposed changes in the attention mechanism.


**Summary Of The Review:**

In summary, the positive points mentioned (clarity of presentation, well-motivated and performant approach, detailed experimental section) clearly outweigh the minor formatting issues. Overall, the proposed approach is a useful and insightful contribution to the field.
This leads to my recommendation to accept the proposed work.

---

### Official Review · Reviewer_AFXh · 2022-11-01

**Confidence:** 4
**Correctness:** 3
**Technical Novelty And Significance:** 3
**Empirical Novelty And Significance:** 3
**Recommendation:** 8

**Clarity, Quality, Novelty And Reproducibility:**

- One distinction between relational transformers and transformer-based GNNs is the inclusion and update of the "global vector". While the authors stated that "many CLRS tasks provide a global feature vector" (Section 5.1.4), I am curious that in general node-level classification tasks (on citation networks and social networks, for example), how should one define the global vectors? In addition, since the objective of relational transformers is to be "a competitive general-purpose model for graph-structured tasks" (Section 1), if global vectors are not provided or are hard to define in certain graph-based tasks, will relational transformers be able to operate on these tasks?

- Following up on the point of "a competitive general-purpose model for graph-structured tasks", though I appreciate the significant performance gains on the CLRS benchmark, it is limited to algorithmic tasks only. From my perspective, node-level, edge-level, and graph-level classifications are also "graph-structured tasks", if not the essential ones. I wonder if the authors could elaborate on why they believe only evaluating algorithmic scenarios would be enough to support the claim of "general-purpose model for graph-structured tasks".

- The authors discussed many graph transformers architectures in Section 4, while most of them were not used as baselines for experiments in Table 1 and beyond. I wonder if it might be a good idea to test out them (GraphFormer and GraphTrans, for example) on the CLRS benchmark, or if there is any technical difficulty that prevented them from becoming baselines in this paper.

- Relational transformers achieved striking performance on "Strings" in Table 1, with 32.52% performance compared to the 1-2% of other baselines. I feel like more explanation and investigation of why RT becomes so much better at handling strings is warranted, if I did not miss it.

- While the performance gain is impressive, I wonder what is the conceptual or empirical advantage of using RT and GNNs instead of classic algorithms on algorithmic tasks.

- There is a ?? in Section 5.1.3.

- Some references are already published, while they are cited as arxiv papers or without any conference/journal information at all. For example, GAT-2 is published at ICLR'22, while there's no conference information in the current reference. Similarly, "Do transformers really perform bad for graph representation" is published at NeurIPS'21, while cited as an Arxiv paper. I hope the authors could fix the references.

- Overall, the authors did a good job discussing related works in GNNs and transformers and positioning this work. Since graph transformers is a growing research area, I suggest the authors include these papers [1,2,3,4,5] in the related work to acknowledge their contribution. Spoiler, one paper proposed "relational graph transformers", which is extremely similar to the name "relational transformers", but very different in content.

[1] Lv, Qingsong, et al. "Are we really making much progress? Revisiting, benchmarking and refining heterogeneous graph neural networks." Proceedings of the 27th ACM SIGKDD Conference on Knowledge Discovery & Data Mining. 2021.

[2] Hu, Ziniu, et al. "Heterogeneous graph transformer." Proceedings of The Web Conference 2020. 2020.

[3] Feng, Shangbin, et al. "Heterogeneity-aware twitter bot detection with relational graph transformers." Proceedings of the AAAI Conference on Artificial Intelligence. Vol. 36. No. 4. 2022.

[4] Hussain, Md Shamim, Mohammed J. Zaki, and Dharmashankar Subramanian. "Global self-attention as a replacement for graph convolution." Proceedings of the 28th ACM SIGKDD Conference on Knowledge Discovery and Data Mining. 2022.

[5] Liu, Xiao, et al. "Mask and Reason: Pre-Training Knowledge Graph Transformers for Complex Logical Queries." Proceedings of the 28th ACM SIGKDD Conference on Knowledge Discovery and Data Mining. 2022.

**Strength And Weaknesses:**

Strengths:
+ interesting research problem of applying GNNs to algorithmic tasks
+ extensive experiments with solid and consistent performance improvements

Weaknesses:
- a few reservations about the experiments and the claim that relational transformers is "a competitive general-purpose model for graph-structured tasks"
- minor issues such as more related work and better presentation

**Summary Of The Paper:**

This paper proposes relational transformers, extending the transformers architecture to general graph-structured data. Specifically, the proposed relational transformers is a graph-to-graph model with feature updates of nodes, edges, and the global graph vector at each layer. Relational transformers achieve remarkable performance on the CLRS benchmark, comprised of algorithmic tasks such as BFS and insertion sort.

**Summary Of The Review:**

Overall, this paper provides technical contributions, is a pleasure to read, and has solid experiments. A few concerns remain in my opinion, and I'm looking forward to discussing them with the authors and revisiting my review and/or score. :)

---

### Official Review · Reviewer_eKg9 · 2022-11-01

**Confidence:** 4
**Correctness:** 3
**Technical Novelty And Significance:** 2
**Empirical Novelty And Significance:** 3
**Recommendation:** 6

**Clarity, Quality, Novelty And Reproducibility:**

The work was clearly written, however there are a few aspects of the presentation (as noted in the weaknesses section) which should be addressed before publication. I felt the overall quality was high, particularly the rigor which the authors spent on the baselines for the primary experiments, however in a few areas (eg. how to handle global vectors, comment on algorithmic analysis) the paper feels a bit incomplete. The work is somewhat novel, and explained well enough to reproduce the architecture from the description in the paper. The authors include the hyperparameters corresponding to the results they report, and included code, so the work should be fully reproducible.

**Strength And Weaknesses:**

### Strengths
1. The idea was pleasingly simple and straightforward, both in motivation and from a technical perspective. The paper is well-structured
- the authors start by listing a loose set of desiderata for any model which adapts the transformers to graphs (including, in particular, updated edge representations while preserving reasonable computational complexity), highlight the ways in which current approaches do not satisfy these objectives, and then present their method which clearly satisfies the requirements.
2. The baselines from the experiments are very rigorous, in the sense that the authors took the time to not only reimplement baselines and verify they obtain similar performance to previously reported results, but then further went on to tune the hyperparameters of these baselines, obtaining better results than previously reported.

### Weaknesses / Questions
1. The standard deviation for RT seems to be much higher than for the other methods, enough so to potentially make the significance of the results and the practical use of RT somewhat questionable. Can the authors provide any insight into why there is more variation with RT than with other methods? In addition, the standard deviation of the baselines also seem to be higher across the board than those reported in the Veličković et al. 2022. Presumably this may be due to the fact that Veličković et al. 2022 only had three evaluations for each, if so then it might be helpful to future readers to explain this difference in the appendix, near where the standard deviations are presented.
2. While much attention (pun intended) was given to the edge vectors, it seems like the global vectors were mostly an afterthought. In particular, the ablation for global vectors compares two methods - concatenation to each input node vector or passing it to a dedicated core node - and finds that the dedicated core node is slightly better (whereas hyperparameter tuning on previous experiments suggested concatenation was better). The authors note that the difference in score between the two was not statistically significant, and suggest that both methods are effective at handling global features, however to make such a claim it would be useful to have an experiment where the global features are not included. Furthermore, while the average score was very similar, the difference per-algorithm seems quite significant. Given the earlier statement about high variability of results, I also think it would be necessary to consider the variance across multiple runs.
3. A further question regarding the global vector - in the concatenation method, does the global vector get updated at each layer? (Obviously the core node approach would update every layer as a result of the node updates.) If not, it seems to run somewhat counter to my understanding of the high-level "pitch" for this paper, i.e. that learning a fully updated attribute graph at every layer was beneficial. I could imagine there are cases where it is useful for the edges *and* nodes to "communicate" shared global information by updating this global representation. While this is indirectly possible with the core node approach, the edges must communicate via updated node representations, which then get passed along additional edges to the core node. The core node (to my understanding of it) is also far from the a "clean" way to handle this, as $2N$ new edges must be added as well along which nodes can communicate with the global vector. A proposal for an approach which handles the global vector in a principled way would make the proposed model and paper more complete.
4. Despite the improved results from properly tuning the baselines, I have concerns that some of the experiments have extremely low numbers, which are the result of an unfortunate choice of hyperparameters. For example, in Table 13 we see that RT-1 performs worse than RT-3, but the results are at least in the same ballpark for most algorithms, whereas for Naive String Match the discrepancy is massive, with RT-1 getting 1.77% whereas RT-3 gets 65.01%. I have similar concerns for the poor performance of the transformer on a few tasks in Table 12.
5. I do not see GAT nor Memnet in Table 11 (baseline reproduction), both of which were in Veličković et al. 2022. I assume the reason Memnet was not included is because it performed worst in Veličković et al. 2022. If this is the case, I think it would be useful to mention it. As for GAT, it seems that sometimes the original model outperformed both GAT-v1 and GAT-v2. This does not change the principal conclusion of the paper, since it does not outperform RT, but I wonder if the authors have any thoughts as to why this is the case. (It should probably be commented on near the reproduction table, at any rate.)
6. The commentary in algorithmic analysis is somewhat weak. The authors attempt to draw some conclusions based on intuition of the model's capabilities and the tasks which it significantly underperforms or overperforms, however the conclusions are simply hypotheses at this point, and would require further analysis and/or experiments to back them up. For example, at the end of this section the authors state "RT can explicitly materialize the DP solution table as its edge vector matrix, and perform updates on each entry using edge updates." which may be true, but does it actually do so?

### Suggestions / Typos
1. The graph-to-graph model formalism described in (1), (2), and (3) does not capture the resulting RT model. Specifically, the edge vector update (2) cannot express the resulting edge vector update (6). If the graph-to-graph model formalism was only intended to abstractly capture the baseline models in section 2, thereby highlighting the distinction of RT, I understand this choice. If not, I just wanted to bring this to your attention.
2. The comprehensive review of the transformer in the appendix is appreciated, however for those equations which are referenced in the main paper it might be worth considering including them in the main text, so that readers will not have to refer to the appendix simply to follow the main discussion. (These include equation 17, 18, and 21.)
3. For notational consistency, it might be nice to replace "nodes" with "vertices", and use $\mathbf v_i$ for the vertex vectors. (Alternatively, one could replace $\mathcal V$ with $\mathcal N$, and all the $v$ subscripts to $n$, to match "nodes", but then the notation for neighbors would need to be changed.) Already, at some points $d_v$ is written as $d_n$ (for example, just under equation (4)).
4. There are a number of places where dimensionality seems incorrect. For example, just under equation (7), I believe it should be $\mathbf W_4 \in \mathbb R^{(2d_e+2d_v) \times d_{eh1}$.
5. In $\mathbf k_{ij}$ and $\mathbf v_{ij}$, the subscript on $\mathbf n$ should be $j$, not $i$.
6. This is very minor, but you can save yourself one line of the equations in section 4 by not distributing the transpose through the matrix-vector products in (11), thereby going directly to (13) and skipping (12).
2. Page 7: missing reference to figure.

**Summary Of The Paper:**

In this work, the authors introduce the relational transformer, a modification of the conventional transformer architecture to work on graphs. This is not the first instance of transformer-inspired architectures which handle graph input, however a fundamental difference is that this method allows for full edge usage, not only incorporating edge attributes when updating node representations, but also allowing edge representations to be updated as well. Each layer essentially takes an attributed graph to another attributed graph, using local information about nodes (i.e. their neighbors and the edge attributes for edges which connect them) to compute a new representation for the node, as well as using local information (i.e. the edge attribute itself, the nodes it connects to, and the reverse edge attribute) to compute a new edge representation. The restriction to local information when computing new edge representations allows the relational transformer to preserve the $\mathcal O(N^2)$ computational complexity of the conventional transformer architecture.

The authors evaluate the new method on the CLRS algorithmic reasoning benchmark. This included the contribution of an extension to this benchmark to allow for models such as relational transformer which update edge representations in each layer, as well as a rigorous strengthening of the baselines by expanding the training datasets and tuning their hyperparameters. The relational transformer is found to outperform these baselines in 12 out of 30, leading to top performance in 6 out of 8 algorithmic classes and scoring highest on average.

**Summary Of The Review:**

While this model proposed by this work performs well empirically, I have some concerns with the high variability of these results (as mentioned in the weaknesses section). I also feel the paper is somewhat incomplete in a few areas at this time, and I think it would benefit from a second revision. As such, I do not feel the paper is ready for publication at this time, but am willing to reconsider my recommendation based on the author's response and discussion with other reviewers.

---

### Public Comment · ~Leslie_O'Bray1 · 2023-02-02
**Correction of the description of SAT (Chen et al. 2022)**

Dear authors,
Thank you for your interesting paper! We wanted to highlight one misunderstanding in the paper: SAT (Chen et al., 2022) uses a GNN inside the transformer layer to generate updated node representations that are structure aware, and is not done prior to the transformer layer as is done e.g. in GraphTrans. SAT uses the GNN in each attention module to update the node embeddings used in the query and key matrices. We would be grateful if you could correct the description in your camera ready version.

---

> ### Author Response · Authors · 2023-02-28
> **Response**
>
> Thanks for bringing this detail to our attention! We have removed our description of SAT as done prior to the transformer.

---

### Decision · Program_Chairs · 2023-01-20

**Decision:**

Accept: notable-top-25%

**Justification For Why Not Higher Score:**

Some parts of the paper can still be improved such as the comparative evaluation with existing graph-based transformers as baselines.

**Justification For Why Not Lower Score:**

The paper is well-motivated and clear-written. Their method is novel and experiment results are persuasive. During the rebuttal, they addressed most concerns and promised to revise the paper accordingly.

**Metareview: Summary, Strengths And Weaknesses:**

This paper presents a straightforward and well-motivated extension of the transformer architecture for the use of graph-structured data, with the proposed model, named the Relational Transformer. The proposed model incorporates relational attention and edge updates, while still retaining the computational efficiency of the original transformer. In their experiments, the proposed model outperformed existing graph neural network baselines on the CLRS benchmark.
The authors carefully address the computational complexity and node-edge communication challenges in existing approaches and demonstrate the effectiveness of their proposed model with empirical evidence. Overall, the paper is well-written and the proposed method looks promising.
The reviewers pointed out room for improvements in this paper such as providing more discussion of other related graph-based transformer models, adding comparative evaluation with those models as additional baselines, and improving the clarity in their description of the use of the global vector.
During the rebuttal, the authors addressed most of the concerns with extensive discussion and revision. They revised the introduction to provide a clearer context for their work in relation to previous research and removed some unsupported claims about how their model compares with other graph-transformer models. They promised to further conduct experiments to compare with other graph-based transformers. After careful review, the AC recommends acceptance.


**Note From Pc:**

if the above contains the word "oral" or "spotlight" please see: "oral" presentation means -> notable-top-5% and "spotlight" means -> notable-top-25%. As stated in our emails, we are disassociating presentation type from AC recommendations